# Circulating Human Serum Metabolites Derived from the Intake of a Saffron Extract (Safr’Inside^TM^) Protect Neurons from Oxidative Stress: Consideration for Depressive Disorders

**DOI:** 10.3390/nu14071511

**Published:** 2022-04-05

**Authors:** Fabien Wauquier, Line Boutin-Wittrant, Line Pourtau, David Gaudout, Benjamin Moras, Adeline Vignault, Camille Monchaux De Oliveira, Julien Gabaston, Carole Vaysse, Karène Bertrand, Hélène Abrous, Lucile Capuron, Nathalie Castanon, David Vauzour, Véronique Roux, Nicolas Macian, Gisèle Pickering, Yohann Wittrant

**Affiliations:** 1Clinic’n’Cell SAS, Faculty of Medicine and Pharmacy, TSA 50400, 28 Place Henri Dunant, CEDEX 1, 63001 Clermont-Ferrand, France; fabien_wauquier@gmx.fr (F.W.); linewittrant@gmail.com (L.B.-W.); 2Activ’Inside, 33750 Beychac et Caillau, France; l.pourtau@activinside.com (L.P.); d.gaudout@activinside.com (D.G.); b.moras@activinside.com (B.M.); a.vignault@activinside.com (A.V.); c.monchaux@activinside.com (C.M.D.O.); julien.gabaston@gmail.com (J.G.); 3Nutrition-Health & Lipid Biochemistry Department, ITERG, 33610 Canejan, France; c.vaysse@iterg.com (C.V.); k.bertrand@iterg.com (K.B.); h.abrous@iterg.com (H.A.); 4INRAE, Nutrition and Integrative Neurobiology (NutriNeuro), UMR 1286, 33076 Bordeaux, France; lucile.capuron@inrae.fr (L.C.); nathalie.castanon@inrae.fr (N.C.); 5Department of Life Science and Health, Nutrition and Integrative Neurobiology (NutriNeuro), Bordeaux University, UMR 1286, 33076 Bordeaux, France; 6Faculty of Medicine and Health Sciences, Biomedical Research Centre, Norwich Medical School, University of East Anglia, Norwich NR4 7TJ, UK; d.vauzour@uea.ac.uk; 7CIC INSERM 1405/Plateforme d’Investigation Clinique CHU Gabriel Montpied, 58 Rue Montalembert, 63000 Clermont-Ferrand, France; v_morel@chu-clermontferrand.fr (V.R.); nmacian@chu-clermontferrand.fr (N.M.); gisele.pickering@uca.fr (G.P.); 8INRAE, UMR 1019, UNH, 63009 Clermont-Ferrand, France; 9Human Nutrition Unit, Clermont Auvergne University, BP 10448, 63000 Clermont-Ferrand, France

**Keywords:** clinical trial, depression, serotonin, BDNF, dopamine, SERT, brain, mental health, crocetin, SH-SY5Y

## Abstract

Increases in oxidative stress have been reported to play a central role in the vulnerability to depression, and antidepressant drugs may reduce increased oxidative stress in patients. Among the plants exerting anti-inflammatory and anti-oxidant properties, saffron, a spice derived from the flower of *Crocus sativus*, is also known for its positive effects on depression, potentially through its SSRI-like properties. However, the molecular mechanisms underlying these effects and their health benefits for humans are currently unclear. Using an original ex vivo clinical approach, we demonstrated for the first time that the circulating human metabolites produced following saffron intake (Safr’Inside^TM^) protect human neurons from oxidative-stress-induced neurotoxicity by preserving cell viability and increasing BNDF production. In particular, the metabolites significantly stimulated both dopamine and serotonin release. In addition, the saffron’s metabolites were also able to protect serotonergic tone by inhibiting the expression of the serotonin transporter SERT and down-regulating serotonin metabolism. Altogether, these data provide new biochemical insights into the mechanisms underlying the beneficial impact of saffron on neuronal viability and activity in humans, in the context of oxidative stress related to depression.

## 1. Introduction

Mood disorders and, in particular, major depressive disorders are among the most prevalent psychiatric disorders and currently represent a leading cause of disability worldwide [1]. Among the mechanisms underlying depression, a growing body of evidence reports a relationship between neuronal alterations and elevated oxidative stress [2]. In addition, it has been reported that stressful life events could also generate a high level of oxidative stress [3,4], which could play a central role in the vulnerability to depression. In agreement with these findings, it has been reported that oxidative stress markers are elevated in patients suffering from major depressive disorders [5]. Those observations were further confirmed in the meta-analysis by Black et al., which reported that oxidative stress measured using 8-OHdG and F2-isoprostanes biomarkers is increased in depression, while antioxidant defenses are altered [6]. The strong relationship between elevated oxidative stress and neurobehavioral alterations has also been demonstrated in animal models. For example, mice treated with L-buthionine-(S,R)-sulfoximine (300 mg/kg), an inducer of oxidative stress, displayed anxiety-like behavioral effects [7]. The pharmacological induction of oxidative stress by xanthine (per os) and xanthine oxidase (i.p.) also caused anxiety-like behavior in rats [8]. The molecular mechanisms underlying these oxidative stress-related anxiety-like behaviors may be related to a decrease in brain-derived neurotrophic factor (BDNF) levels and, interestingly, the depression-like phenotype may be reversed by treatment with antioxidants [9]. Remarkably, selective serotonin reuptake inhibitor (SSRI) antidepressants have been reported to increase serum concentrations of BDNF in patients suffering from major depressive disorder and contribute to improvements in the symptoms [10]. Furthermore, antidepressant drugs have antioxidant properties and reduce increased oxidative stress in patients [11]. Therefore, elucidating the effects of antidepressants on oxidative stress represents an important target to understand the mechanism of antidepressant treatment.

Although a large variety of antidepressant medications is currently available, a significant proportion of patients does not respond to them adequately, highlighting the need to continue to search for alternative solutions. Saffron, a spice obtained from the stigmas of *Crocus sativus* L., a Mediterranean plant belonging to the Iridaceae family, is used as a highly valued plant for traditional medicinal purposes [12] and, in particular, for its positive effects on mood and depression [13,14,15,16,17]. Saffron stigmas are rich in crocins and crocetins, two carotenoid pigments responsible for its color; in picrocrocin, which is responsible for its flavor and bitter taste; and in safranal, a volatile compound responsible for its aroma and smell. In addition to their organoleptic properties, these bioactive components also show anti-inflammatory and anti-oxidant properties [18,19,20] and have been reported to exert health-related effects on emotional well-being [13,14,15,16,21]. At a cellular level, saffron was proven to regulate apoptosis, modulate inflammatory pathways, and affect mitochondrial dysfunction by upregulating gene expression, including Cyr61, Gpx8, Ndufs4, and Nos1ap, which possess neuroprotective properties [22]. The beneficial effects of saffron on depression have mainly been associated with its potential SSRI-like activity, the classical mechanism of antidepressant medications [14]. Accordingly, we recently demonstrated that the behavioral improvements following Safr’Inside^TM^ administration were associated with a positive modulation of both serotonergic and dopaminergic neurotransmission in naive mice [23]. However, the exact mechanisms underlying antidepressant neuroprotection in the context of oxidative stress have not been completely elucidated. To address this knowledge gap and contribute to the determination of saffron’s mechanisms of action in humans, we designed an ex vivo clinical protocol based on human serum enrichment. The robustness of this pioneering approach was recently validated and published [24,25,26,27]. In this study, we combined human metabolism and cell cultures to try to understand whether and how circulating metabolites resulting from saffron extract consumption in humans may influence the activity of human neurons in the context of the oxidative stress commonly found in depression.

## 2. Materials and Methods

### 2.1. Ethics Clinical Trial

This study was conducted in accordance with the Declaration of Helsinki of 1975 (https://www.wma.net/what-we-do/medical-ethics/declaration-of-helsinki/, accessed on 1 April 2021), revised in 2013. The human study was approved by the French Ethical Committee (2021T2-02 RIPH2 HPS/N° SI RIPH: 21.01.11.58647/N° EudraCT/ID RCB: 2020-A3184-35/Comité de Protection des Personnes CPP Tours-Région Centre-Ouest I; approved 11 March 2021). The volunteers were informed of the objectives and the potential risks of the present study and provided their written informed consent before they participated in the study.

### 2.2. Human Study Design and Pharmacokinetics of Absorption

A pool of 10 healthy men (age: 25.0 years old, +/−5.1; BMI: 23.9 kg/m^2^, +/−2.3; >60 kg; without drug treatment; and no distinction with regards to ethnicity) volunteered for this study. They were tested for normal blood formulation and renal (urea and creatinine) and liver functions, including aspartate aminotransferase (AST), alanine aminotransferase (ALT), and gamma-glutamyltransferase (GGT) activities. Blood samples from all participants were obtained and collected in serum-separating tubes. Serum was prepared, aliquoted, and stored at the Centre d’Investigation Clinique de Clermont-Ferrand—Inserm 1405, a specialised research department that guarantees the quality of samples and compliance with regulatory and ethical obligations (certification according to the French standard NF S 96900).

The first step of the study aimed at determining saffron extract’s metabolite absorption peak, and especially crocetins. Ten healthy volunteers who fasted for 12 h were given 300 mg of a saffron extract (SaE). The dose was set according to validated preclinical [28,29,30] and clinical data [14,21,31,32,33,34,35]. The SaE was given as one capsule. Approximately 9 mL of venous blood was collected from the median cubital vein before the ingestion and at different time points after the ingestion (5, 10, 15, 30, 45, 90, 120, 150, 180, 210, 240, and 300 min). Serum was prepared from venous blood samples and stored at −80 °C until analysis. Crocetin absorption profile was evaluated by ultra-high-performance liquid chromatography (UHPLC) coupled mass spectrometry. Briefly, for identification and quantification of serum crocetins, samples were first treated with acetonitrile to precipitate the proteins and then saponified, extracted with ethyl acetate, and analysed by ultra-high-performance liquid chromatography tandem mass spectrometry (UHPLC-MS/MS) (ThermoFisher Scientific, Courtaboeuf, France), according to the modified chromatographic described by Zhang et al. [36]. Once the absorption peak was determined, volunteers were called back for the collection of the enriched serum fraction. For this second clinical phase, eight healthy volunteers who fasted for 12 h were given 300 mg of the SaE as one capsule. Approximately 48 mL of venous blood was drawn from the cubital vein before the ingestion for the collection of a naïve serum. Next, at the maximum absorption peak, 48 mL of blood was drawn for enriched serum collection. Serum was stored at −80 °C until analysis.

### 2.3. Saffron Extract (SaE)

The saffron extract (Safr’Inside^TM^; Activ’Inside, Beychac-et-Caillau, France) was a full-spectrum standardised extract obtained according to the patent FR 3054443 - WO2017EP69200. This extract contained crocins (mainly *trans*-4-GG, *trans*-3-Gg; *cis*-4-GG, *trans*-2-G) >3%, safranal >0.2%, picrocrocin derivatives (mainly picrocrocin, HTCC) >1%, and kaempferol derivatives (mainly kaempferol-3-sophoroside-7-glucoside, kaempferol-3-sophoroside) >0.1%, measured by UHPLC method.

### 2.4. Human Neuron Cultures

The human neuroblast cell line SH-SY5Y was purchased from Abcam^®^ (Paris, France, ab275475). The SH-SY5Y cells were cultured in MEM (Minimum Essential Medium Eagle, Sigma-Aldrich, Lyon, France, M0325)/F12 (F12 nutrient mixture, Sigma-Aldrich, N3520) medium (1:1) supplemented with 0.5% (*v*/*v*) non-essential amino acid solution (Sigma-Aldrich, M7145), 0.5% (*v*/*v*) L-glutamine solution (Sigma-Aldrich, G7513), and 1% penicillin/streptomycin (Life Technologies, Villebon-Sur-Yvette, France). For maintenance, the medium was also supplemented with 10% FCS (Invitrogen). Cells were grown at 37 °C in an atmosphere of 5% CO_2_/95% air in either in 96- or 24-well plates with 100 µL or 500 µL of culture medium, respectively. To induce neuronal differentiation, cells were plated at 12,000 cells/cm^2^ in the MEM/F12 (1:1) medium supplemented with 2.5% FCS and 10 µM RA (all-trans retinoic acid, Sigma-Aldrich, R2625). After five days, the medium was changed to a MEM/F12 (1:1) medium supplemented with 2.5% FCS, 10 µM RA, and 50 ng/mL BDNF (brain-derived neurotrophic factor, Sigma-Aldrich, B3795). Cells were then allowed to differentiate for four more days before further experiments were carried out.

To analyse the effects of the SaE serum metabolites on neurotoxic stress, differentiated SH-SY5Y cells were preincubated for 24 h in the MEM/F12 (1:1) medium in the presence of 10% of human naïve serum or human enriched serum, according to the Clinic’n’Cell protocol (DIRV INRA 18-00058), prior to an additional 24-hour treatment with 500 µM H_2_O_2_ (Sigma-Aldrich, H1009).

### 2.5. Cell Viability

The ex vivo cell viability was determined using an XTT-based method (Cell Proliferation Kit II, Sigma-Aldrich) according to the supplier’s recommendations. Optical density was measured at 450 nm.

### 2.6. Reactive Oxygen Species (ROS) Measurement

SH-SY5Y were seeded on 96-well dark-wall clear-bottom plate at 12,000 cells/cm^2^. Twenty-four hours following H_2_O_2_ stimulation, cells were washed and incubated with 10 µM of 2′,7′-dichlorofluorescin diacetate (DCF-DA) solution (ab113851, Abcam) for 45 min at 37 °C in the dark, and rinsed with the dilution buffer according to the manufacturer’s protocol. Fluorescence was then measured with a fluorescence plate reader (Berthold − Mitras) at Ex/Em = 485/535 nm in end-point mode. 

### 2.7. Brain-Derived Neurotrophic (BDNF) Factor, Serotonin, 5-Hydroxyindoleacetic Acid (5-HIAA), and Dopamine Quantification

In SH-SY5Y cells, BDNF levels were evaluated in cell culture supernatant using the Human BDNF ELISA Kit from Abcam (ab212166), following the manufacturer’s instructions.

Serotonin levels were evaluated in cell culture supernatant using the serotonin ELISA Kit from Abcam (ab133053), according to the manufacturer’s recommendations., A metabolite from cellular degradation of serotonin, 5-HIAA, was measured in SH-SY5Y cell culture supernatant using the 5-HIAA ELISA kit from Immusmol SAS (Bordeaux, France, BA-E-1900). Briefly, 5-HIAA was first derivatised by methylation. The quantification of the newly formed methylated analyte was then performed by competitive ELISA. The reaction was monitored at 450 nm. 

Dopamine was measured in SH-SY5Y cell culture supernatant using the dopamine ELISA kit from Immusmol SAS (BA-E-5300). Briefly, dopamine was first extracted by using a cis-diol-specific affinity gel. After its acylation and enzymatic conversion, the quantification of this derivatised dopamine was achieved by competitive ELISA. The reaction was monitored at 450 nm. 

### 2.8. Real-Time RT-qPCR

The mRNAs from differentiated SH-SY5Y cells were isolated using TRIzol™ Reagent (Ambion–Life Technologies), according to the supplier’s recommendations. Serotonin reuptake transporter (SERT) mRNA expression levels were measured by RT-qPCR (PowerUp SYBRgreen, Applied Biosystems). β-Actine was used as a housekeeping gene. Primers were designed as follows: SERT-F: 3′-GGC TCA TCA GAA AAC TGC AAA-5′; SERT-R: 3′-GCT GTG TCT TGG TTC TAT GGC-5′; ACTβ-F: 3′-ATT GGC AAT GAG CGG TTC-5′; and ACTβ-R: 3′-GGA TGC CAC AGG ACT CCA5′.

### 2.9. Statistics

Results are expressed as mean +/− SEM (standard error of the mean). Statistical analyses were carried out using ExcelStat Pro (Microsoft, Issy-les-Moulineaux, France). One-way ANOVA was performed, followed by Fisher post hoc test. Global *p*-value and F-value are indicated in the figure legend for each ANOVA of each analysed parameter. Groups significantly different from each other (*p* < 0.05) are indicated with different letters. Groups with no significant statistical difference from each other share the same letter. 

## 3. Results

### 3.1. Kinetic Profile of the SaE Absorption 

The clinical study was carried out in two phases. The first phase aimed at characterising the metabolites present in human serum following consumption of a standardised SaE in order to determine the time frame of their absorption peak. The second phase was dedicated to collecting both naïve and enriched sera, before ingestion and at the absorption peak, respectively. Once the blood samples were collected, we determined the influence of the sera enriched in metabolites of interest resulting from the consumption of the SaE on the function of the human neurons.

To track the metabolites’ enrichment in serum, we used crocetin as an accurate and validated marker of SaE absorption. Crocins were found in the studied extract and most of them were converted into crocetins during the digestion/absorption processes. The fasted volunteers received 300 mg of SaE and the absorption profile of the crocetins was monitored for 5 h. The analysis of the absorption profile showed that the maximum crocetin concentration was reached 90 min post-ingestion (data not shown). Consequently, enriched serum with SaE metabolites was collected at 90 min post-ingestion for the second phase of the clinical protocol.

### 3.2. Validation of the Neuronal Differentiation and Ex Vivo Procedures

The aim of this study was to evaluate the impact of circulating metabolites following SaE ingestion on mature human neuron homeostasis. In order to obtain a homogenous cellular population of mature neurons, the cell differentiation was driven by retinoic acid and BDNF (Figure 1A). However, in their undifferentiated state, the SH-SY5Y grew rapidly, appeared to be non-polarized, and had very few, and short, processes (Figure 1B(a)). After 9 days, cells undergoing differentiation showed a reduced proliferation rate associated with a polarisation characterised by the presence of long neurite-like processes (Figure 1B(b)). These morphological changes were indicative of effective neuronal maturation, and all the cellular investigations were conducted at this stage. 

Next, to ensure the relevance of our ex vivo approach, we checked the influence of the different human sera on cell growth and we compared it to regular fetal calf serum incubation by measuring the XTT-based activity. Consistently, cell growth stopped in the serum-free cultures, while it was maintained in the presence of FCS 10% (Figure 1C). The naïve or enriched human sera processed according to the Clinic’n’Cell methodology (DIRV#18-0058; see the Patents section) did not exert any adverse effects on the cells compared to the conventional fetal calf serum and allowed cell growth and viability similar to the 10% FCS (Figure 1C). In addition to growth/viability, the baseline influence of the human sera on the other parameters (ROS Production, BDNF level, serotonin and 5HIAA metabolite level, and dopamine level) of the study were also checked with regards to the control FCS (Appendix A). These data validated the performance of further ex vivo investigations.

### 3.3. Human Serum Containing Metabolites from SaE Preserves Human Neurons from Oxidative-Stress-Induced Neurotoxicity

As expected, the treatment with 500 µM H_2_O_2_ caused cellular damages to the differentiated neurons, including reduced viability (Figure 2A) and increased intracellular levels of ROS (Figure 2B). The presence of the SaE-enriched serum fully inhibited the H_2_O_2_-related modulation of both parameters (Figure 2A,B). BDNF is an important neurotrophic factor. The inhibition of BDNF production by H_2_O_2_ treatment was described previously [9,37]. Here, we were able to reproduce these data in the presence of naïve human serum. Remarkably, the metabolites from the SaE promoted the basal production of BDNF and counteracted the deleterious effect of the H_2_O_2_ treatment on BDNF production (Figure 2C). Altogether, the SH-SY5Y cells incubated with sera enriched with metabolites from SaE ingestion were protected from H_2_O_2_-induced neurotoxicity.

### 3.4. Human Serum Containing Metabolites from SaE Enhance Basal Activity of Serotonergic System and Protect It from Oxidative-Stress-Induced Alterations

The dysregulation of the serotonergic system is commonly associated with depressive disorders, and many antidepressants rely on their ability to inhibit serotonin recapture through the SERT, leading to higher serotonin levels in the synapse [38,39]. Therefore, we investigated serotonin release and serotonin transporter expression in our cellular model of mature neurons. 

The H_2_O_2_ treatment led to a significant reduction in serotonin release by the differentiated SH-SY5Y cells (Figure 3A), which was associated with an increase in SERT mRNA expression (Figure 3C). The extracellular levels of 5-HIAA, a product of serotonin degradation, were also significantly increased by the H_2_O_2_ treatment (Figure 3B). Overall, we showed that the serotonergic system was significantly dysregulated in our cellular model under oxidative stress. 

The presence of SaE-enriched serum significantly inhibited the H_2_O_2_-induced dysregulation of the serotonergic parameters, thus normalising the serotonin release (Figure 3A), SERT mRNA expression (Figure 3B), and 5-HIAA levels (Figure 3C). These data suggest that the presence of metabolites from SaE ingestion reduces both serotonin recapture and degradation under oxidative stress conditions, leading to the preservation of the serotonergic system. 

Interestingly, even when the cells were not challenged with H_2_O_2_, the serum enrichment significantly increased the serotonin release (Figure 3A) and down-regulated the SERT mRNA expression (Figure 3B), further supporting the brain-health benefits of saffron’s metabolites.

### 3.5. Human Serum Containing Metabolites from SaE Stimulates Dopamine Production

Depression includes the dysregulation of the dopaminergic system [23,40]. Thus, we checked the dopamine release in our human neuron cultures. Interestingly, the presence of circulating metabolites from the SaE significantly and potently enhanced the dopamine release (Figure 4), regardless of the presence or absence of oxidative stress. 

## 4. Discussion

Using an original ex vivo clinical approach, we demonstrated that the circulating metabolites produced following SaE intake in humans protect human neurons from oxidative-stress-induced neurotoxicity by preserving cell viability and BNDF production, while blunting ROS production. In these conditions of oxidative stress, saffron’s metabolites also stimulate both dopamine and serotonin production. Furthermore, they inhibit SERT expression and down-regulate serotonin metabolism. Altogether, this innovative study provides valuable new insights into the mechanisms that are likely to contribute to the beneficial impact of saffron on human brain function. 

To obtain the greatest enrichment of the saffron extract’s metabolites in human serum, we investigated the pharmacokinetic profile of the saffron extract’s absorption. To achieve this aim, crocetin measurement remains the most accurate and validated approach through which to investigate the absorption of crocin derivatives in plasma or serum matrices. Based on the crocetin absorption, we set the serum collection at 90 min post-ingestion. Notably, a previous clinical study showed that crocetin could peak at around 4 h post-ingestion [34]. More recently, Almodovar et al. showed a maximum crocetin absorption between 60 min and 90 min after saffron extract ingestion and suggested that the kinetics may be dependent on the dose [31]. Nevertheless, these clinical designs were set up differently. Thus, it remains difficult to further compare the different studies; the fasted state or the meal association may account for these apparent discrepancies. 

The recommended daily intake of saffron extract is around 30 mg/day. In this study, the enriched sera were diluted ten times for cell culture investigations; therefore, we aimed at setting the single exposure of saffron extract at a dose of 300 mg for more accurate nutritional consistency. To keep this acute exposure safe and physiologically sound for this ex vivo clinical protocol, the dose was compared to the literature data. Most of the clinical trials on saffron extracts use a dose comprising between 30 and 100 mg/day for an exposure period ranging from 2 to 12 weeks [14]. The main reported side effects of saffron overconsumption are related to bleeding and vascular dysfunction. Interestingly, two clinical studies investigated doses of 200 and 400 mg/day for 7 days, and both studies concluded that there was an absence of adverse side effects [32,33]. Here we used a single dose of 300 mg, and none of the volunteers reported any adverse events. 

This human ex vivo approach considers the systemic aspect of digestion, particularly the intestinal barrier and hepatic transformation. Indeed, crocins are converted into crocetins in the intestine by enzymatic processes in the epithelial cells [41]. This integrative methodology also makes it possible to test the whole metabolome found in serum after full-spectrum saffron extract ingestion. Consequently, we keep intact the potential nutritional synergies that may occur at the whole-body level. However, the digestive track may not be the only border that needs to be crossed in vivo, and saffron’s benefits for brain health may also rely, at least in part, on the ability of its blood metabolites to enter the brain. In support of our approach, crocetins were recently reported to cross the blood–brain barrier and reach the central nervous system by passive transcellular diffusion [41,42]. Furthermore, crocetins may also enter the brain regardless of their method of administration, whether by intraperitoneal or intravenous injection [43] or by oral ingestion [41], further supporting the physiological relevance of this clinical ex vivo approach.

Oxidative stress is a commonality in the pathophysiology of neurodegenerative disorders, such as Alzheimer’s disease, Parkinson’s disease, Huntington’s disease, amyotrophic lateral sclerosis and multiple sclerosis [44], anxiety, and depression [6,7,8,9,45,46]. The pathogenesis of anxiety and depression has been linked to oxidative stress via alterations in antioxidant defenses [6] and cyclic nucleotide signaling [7]. These preclinical and clinical data strongly suggest that anti-oxidant strategies, including nutritional strategies, may represent a potent alternative to pharmacological antidepressant treatments by targeting the neurotransmission system to improve depression management [46]. For instance, in rodent models, the development of anxiety-like behaviors was blunted by apocynin, a NADPH oxidase inhibitor [7]. In addition, both antioxidants and treadmill exercise training reduced oxidative stress in the rat brain regions implicated in anxiety response and prevented oxidative-stress-induced anxiety-like behavior [45]. 

A persistent state of oxidative stress has been associated with both vulnerability to depression and low brain-derived neurotrophic factor (BDNF) levels [9]. In a rat model, heat stress of the hippocampus induced oxidative damage and perturbation in the BDNF/ERK1/2/CREB pathway in brain tissues [37]. Interestingly, BDNF constitutively controls the nuclear translocation of the master redox-sensitive transcription factor Nrf2, which activates antioxidant defenses [9], highlighting the inter-relationships between BDNF, oxidative stress, and depression. Consistently, we found that H_2_O_2_ drives both oxidative stress and the downregulation of BDNF production in human neurons. More importantly, this alteration in neuronal activity is prevented in the presence of the circulating metabolites of saffron. Along with oxidative stress impairment, the pathophysiology of depression may include glucocorticoids excess, and corticosterone is an acknowledged inducer of anxiety and depression models [46,47,48]. Interestingly, we found that corticosterone induced oxidative stress in human neurons, further suggesting the presence of a central oxidative mechanism (Appendix A).

In this study, we also questioned whether saffron’s antioxidant properties may act in concert with its previously described SSRI-like capabilities. Interestingly, evidence is accumulating on the influence of oxidant status upon the regulation of the serotonergic system and vice-versa. Khanzode et al. suggested the potential antioxidant action of selective serotonin re-uptake inhibitors at a clinical level [39]. More recently, SSRIs found to prevent oxidative-stress-related DNA damage in patients with depression [38]. 

From a mechanistic point of view, oxidative environments increase both the expression of the serotonin transporter (SERT) and the uptake of serotonin prior to its degradation into 5-hydroxy-indolylacetic acid (5-HIAA) [49]. By contrast, SSRIs such as fluoxetine inhibit the oxidative-stress-induced expression of SERT in rats [50]. The anti-oxidative properties of rooibos tea (*Aspalathus linearis*) reversed the oxidative stress-related increase in 5-HIAA in rat brains in a model of immobilization stress [51]. Together, these mechanisms drive the increase in serotonin levels, which has been demonstrated in turn to inhibit oxidative-stress-related neurotoxicity in HT-22 mouse neuronal cells through the activation of both TrkB/CREB/BDNF and Nrf2 pathways [52]. Saffron is known for its health benefits, potentially through its SSRI-like properties, but the evidence of this mechanism remained to be documented. In this study, we report that the human circulating metabolites resulting from full-spectrum saffron extract intake enhanced basal serotonin levels and maintained normal SERT expression and 5-HIAA levels upon oxidative stress insult, thereby providing insights into how saffron extract, and most notably Safr’Inside^TM^, preserves serotonergic tone in humans. 

Furthermore, saffron was very recently reported in clinic as synergistically increasing both serotonin and dopamine levels in serum when combined with endurance exercise in young adults [40]. In agreement with these data, we consistently found that the circulating metabolites from saffron were able to stimulate both serotonin and dopamine production in human neurons in both the presence and the absence of oxidative stress. 

The data reported in this ex vivo clinical study are fully consistent with our previously published preclinical results and further support the brain benefits of saffron. Indeed, we showed that the reduced depressive-like behavior of naive mice in a forced swim test following Safr’Inside^TM^ administration was associated with a positive modulation of both serotonergic and dopaminergic neurotransmission [23]. This supplementation with Safr’Inside™ led to similar anti-depressant effects to those achieved with SSRI or pure safranal administration, which could be attributed to the measured reduction in SERT expression in the same mice. 

Here, we provide details, at a clinical level, on the role and the mode of action of full-spectrum saffron’s metabolites regarding the preservation of neuron activity in the context of oxidative stress, which is commonly found in depression. To the best of our knowledge, such relationships have not yet been reported.

## 5. Conclusions

Using a pioneering clinical ex vivo approach, considering the digestive processes of nutrients, we presented clues as to the role of the circulating metabolites produced following saffron extract intake in humans. Furthermore, we provided further biological and clinical evidence regarding the mode of action driving the beneficial health effects of full-spectrum saffron extract on depression (Figure 5).

## 6. Patents

The human ex vivo methodology used in this study has been registered as a written invention disclosure by the French National Institute for Agronomic, Food and Environment Research (INRAE) (DIRV#18-0058). Clinic’n’Cell^®^ has been registered as a mark [24,25,26,27].

## Figures and Tables

**Figure 1 nutrients-14-01511-f001:**
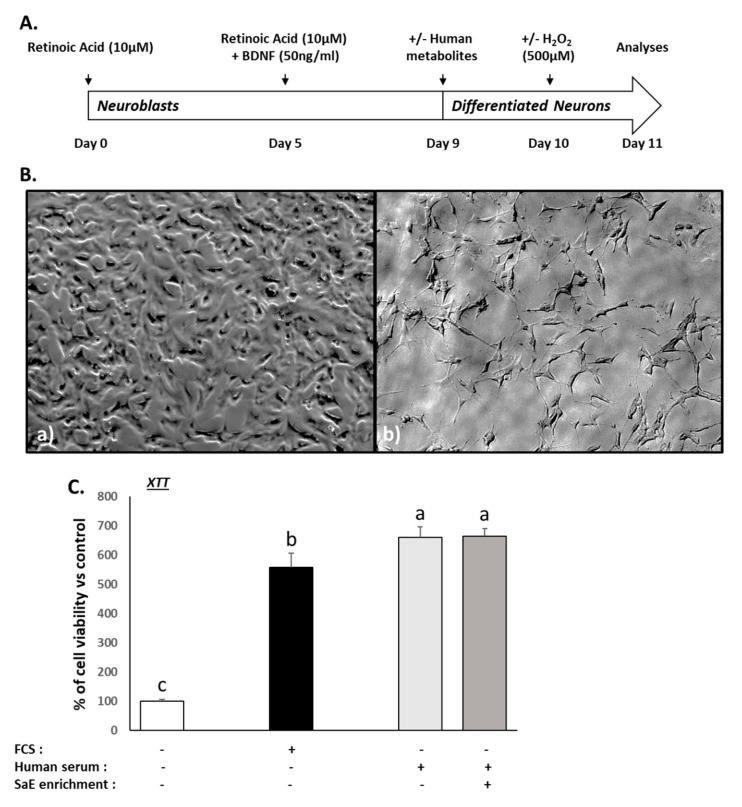
SH-SY5Y cells’ differentiation into mature neurons and impact of human serum enriched with SaE, or not, on cellular viability. The differentiation protocol using retinoic acid and BDNF (**A**) led to progressive morphological changes, including neurite growth (**B**(**b**)), compared to undifferentiated cells (**B**(**a**)). Compared to FCS, cellular viability was slightly increased in SH-SY5Y cells exposed to human serum for 48 h, as measured with an XTT-based assay (**C**). Measures were performed in quadruplicates per condition/volunteer (*n* = 8 volunteers). Groups significantly different from each other (*p* < 0.05) are indicated with different letters. Groups with no significant statistical difference from each other share the same letter. XTT: global ANOVA *p*-value < 0.001 and *F*-value: 210.351.

**Figure 2 nutrients-14-01511-f002:**
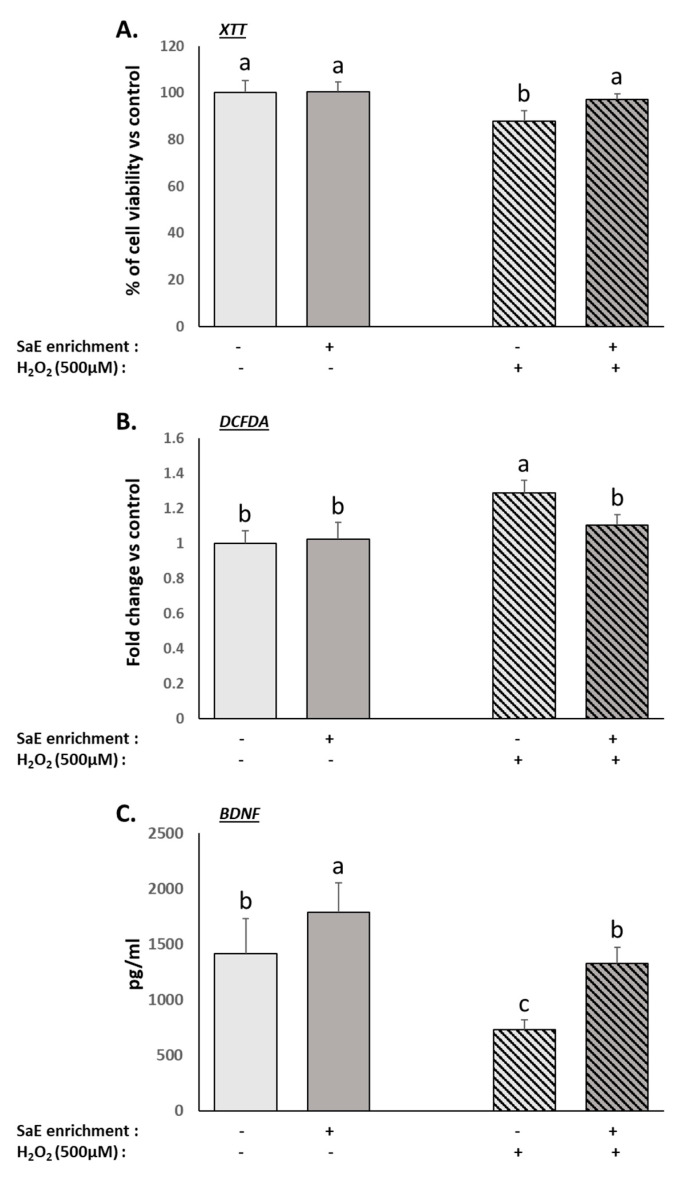
Effect of human serum enriched with SaE metabolites on the cellular injuries caused by H_2_O_2_-induced oxidative stress in SH-SY5Y cells. On day 9, differentiated SH-SY5Y cells were incubated with human serum for 24 h prior to an additional 24 h of treatment with 500 µM H_2_O_2_. Proliferation was measured using an XTT-based assay (**A**). Intracellular ROS levels (**B**), as well as BDNF production (**C**), were measured using DCFDA- and ELISA-based assays, respectively. SaE enrichment was able to significantly counteract the decreased viability (**A**) and BDNF production (**C**), as well as the increase in intracellular ROS (**B**) caused by H_2_O_2_ treatment. Measurements were performed in quadruplicates for each sample of the eight volunteers. Groups significantly different from each other (*p* < 0.05) are indicated with different letters. Groups with no significant statistical difference from each other share the same letter. XTT: global ANOVA *p*-value < 0.001 and *F*-value: 6.592. DCFDA: global ANOVA *p*-value < 0.001 and *F*-value: 9.536. BDNF: global ANOVA *p*-value < 0.001 and *F*-value: 27.722.

**Figure 3 nutrients-14-01511-f003:**
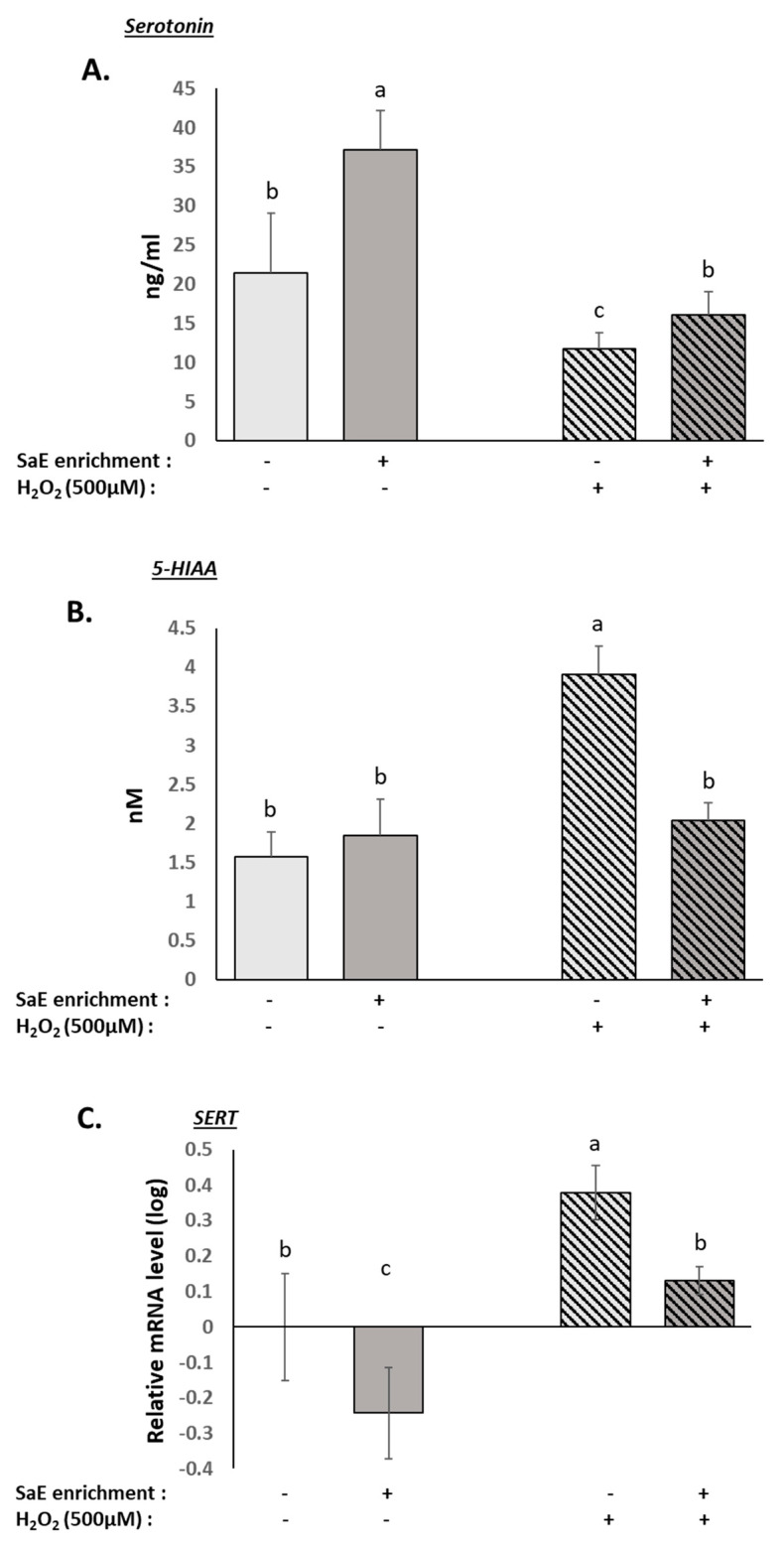
Effect of human serum enriched with SaE metabolites on the serotonin pathway dysregulations caused by H_2_O_2_-induced oxidative stress in SH-SY5Y cells. Differentiated SH-SY5Y cells were incubated with human serum for 24 h prior to an additional 24 h of treatment with 500 µM H_2_O_2_. Serotonin (**A**) and 5-HIAA (**B**) production was measured using ELISA-based assays. SERT mRNA expression was analysed by RT-qPCR and expressed in log^10^ scale relative to the control SaE(−)/H_2_O_2_(−) (**C**). H_2_O_2_ inhibited serotonin release (**A**) and increased both 5HIAA (**B**) production and SERT mRNA expression (**C**). SaE enrichment led to a restoration of serotonin pathway homeostasis. Measurements were performed in quadruplicates for each sample of the eight volunteers. Groups significantly different from each other (*p* < 0.05) are indicated with different letters. Groups with no significant statistical difference from each other share the same letter. Serotonin: global ANOVA *p*-value < 0.001 and *F*-value: 24.683. 5-HIAA: global ANOVA *p*-value < 0.001 and *F*-value: 38.620. SERT: global ANOVA *p*-value < 0.001 and *F*-value: 29.500.

**Figure 4 nutrients-14-01511-f004:**
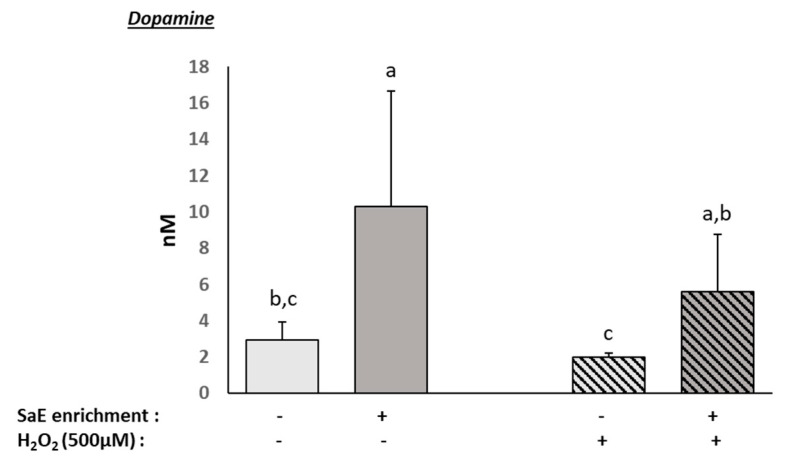
Effect of human serum enriched with SaE metabolites on the dopamine release by SH-SY5Y cells in oxidative stress conditions. Differentiated SH-SY5Y cells were incubated with calf serum or human serum for 24 h prior to an additional 24 h of treatment with 500 µM H_2_O_2_. Extracellular dopamine was measured using an ELISA-based assay. H_2_O_2_ induced a reduction in dopamine release, which was counteracted by the presence of SaE-enriched serum. Measurements were performed in quadruplicates for each sample of the eight volunteers. Groups significantly different from each other (*p* < 0.05) are indicated with different letters. Groups with no significant statistical difference from each other share the same letter. Dopamine: global ANOVA *p*-value = 0.010 and *F*-value: 4.043.

**Figure 5 nutrients-14-01511-f005:**
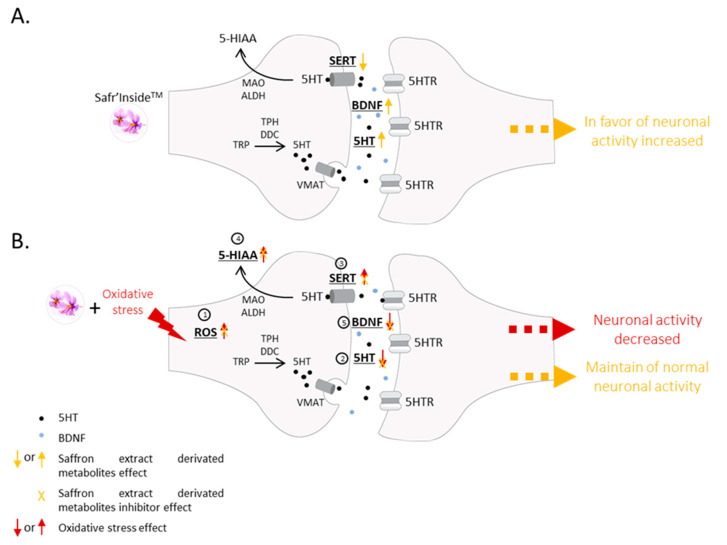
Schematic representation of saffron extract effects, oxidative stress effects, and potential protective effects of saffron extract on neurons. (**A**): In the absence of oxidative stress conditions, saffron extract metabolites promote serotonin and BDNF release, as well as the down-regulation of serotonin reuptake transporter (SERT) expression. (**B**): Oxidative stress is known to induce intracellular reactive oxygen species (ROS) production (1). In serotoninergic neurons, oxidative stress is associated with lower serotonin release (2) and higher (SERT) expression (3). Consistent with this SERT up-regulation, serotonin catabolism is enhanced and levels of 5-Hydroxyindoleacetic acid (5HIAA) metabolite rise (4). Together, these altered serotonin parameters support decreased serotonergic neurotransmission. In addition, brain-derived neurotrophic factor (BDNF) production is impaired under oxidative stress conditions (5). Promotion of serotonin release in absence of oxidative stress conditions contributes to the prevention of oxidative-stress-related alterations in the serotonin system and the preservation of normal serotonergic neurotransmission under stress. Saffron extract metabolites inhibit both ROS production and oxidative-stress-related BDNF decrease, thereby participating in the preservation of neuron activity as well. TRP: Tryptophan. TPH: Tryptophane hydroxylase. DDC: 5-hydroxytryptophan decarboxylase. 5HT: Serotonin. MAO: Monoamine oxidase. ALDH: Aldehyde dehydrogenase. 5HTR: Serotonin receptor.

## Data Availability

The data presented in this study are available on request from the corresponding author. The data are not publicly available due to ethical restrictions.

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
