# Peer review of "Circulating Human Serum Metabolites Derived from the Intake of a Saffron Extract (Safr’InsideTM) Protect Neurons from Oxidative Stress: Consideration for Depressive Disorders"

_nutrients, 2022, doi:10.3390/nu14071511_

Round 1

Reviewer 1 Report

The Manuscript: ,, Circulating human serum metabolites derived from intake of saffron extract  (Safr’insideTM) protect neurons from oxidative stress: consideration for depressive disorders'' written by Wauquier et al., is very interesting orginal paper that analyses the pharmacokinetic metabolism in vivo as well as influence of saffron metabolites on human neurons ex vivo. Generally the results are very interesting. It is analysed both the the influence of saffron metabolites on the oxydative stres, BDNF production and SERT expression and serotonin metabolism.

I have several remarks concerning the Materials and Methods and Resuls sessions:

Could you explain why in the first phase of Human Study take part 10 healthy men while in the second only 8. 

I suggest to describe more detaily results, please provide the F value for ANOVA, please decribe what exactly mean significant differences indicated by a,b, c and d letters. I could not found these informations in the decription of the figure as well as in the text.

Author Response

Reviewer 1

The Manuscript: “Circulating human serum metabolites derived from intake of saffron extract  (Safr’insideTM) protect neurons from oxidative stress: consideration for depressive disorders'' written by Wauquier et al., is very interesting original paper that analyses the pharmacokinetic metabolism in vivo as well as influence of saffron metabolites on human neurons ex vivo. Generally the results are very interesting. It is analysed both the influence of saffron metabolites on the oxydative stress, BDNF production and SERT expression and serotonin metabolism.

We thank you for such comments and compliments regarding our work, its interest and originality.

I have several remarks concerning the Materials and Methods and Results sections:

Could you explain why in the first phase of Human Study take part 10 healthy men while in the second only 8. 

We had 2 dropouts between the two phases due to the health context. Since we had to respect and follow the timeline of the protocol according to ethical policies, thus we initiated the second phase with 8 volunteers instead of 10. 

I suggest to describe more detaily results, please provide the F value for ANOVA, please decribe what exactly mean significant differences indicated by a,b, c and d letters. I could not found these informations in the decription of the figure as well as in the text.

We agree, and we added the global p-value and F-value in the figures’ legend for each ANOVA of each analyzed parameter. Regarding the letters (a, b, c, d…), they were tagged to groups according to p-value for each group comparison. Significance was set to p<0.05 for each group comparison. To improve clarity, we have changed the sentence to “Groups significantly different from each other (p<0.05) are indicated with different letters. Groups with no significant statistical difference from each other share the same letter.” We also included this sentence in the supplemental material.

Reviewer 2 Report

Review of a manuscript “Circulating human serum metabolites derived from the intake of a saffron extract (Safr’insideTM) protect neurons from oxidative stress: consideration for depressive disorders” by Fabien Wauquier and coauthors submitted top “Nutrients” MDPI.

A growing number of mood disorders, including major depressive disorders are a large group of diseases often leading to disability. A significant role in the development of these disorders is played by damage to neurons caused by oxidative stress.  Increased level of oxidative stress is often associated with neurobehavioral alterations, however, the exact mechanism linking them is not completely understood.

The authors developed a robust ex vivo clinical protocol based on human serum enrichment aiming to reveal the connection between activity of human neurons, oxidative stress in depression and the effect of saffron extract on these processes. This is an important area of biomedical research, and the results presented in the manuscript will be interesting for the readership of “Nutrients”.

The following corrections should be made.

Abstract

Line 42-43: “Altogether, these data provide new clinical insights into the mechanisms underlying the beneficial impact of saffron on neuronal viability and activity”. The data presented in the manuscript provide more biochemical, than clinical insights into the mechanisms underlying the beneficial impact of saffron”

Introduction

Lines 83-84: “…and have been reported to promote health-related benefits on emotional well-being [13-16, 21]” . After this sentence the authors should add the following sentence and citation: ”Saffron regulates apoptosis, modulates inflammation and affects mitochondrial dysfunction by upregulating genes, for example, Cyr61, Gpx8, Ndufs4, and Nos1ap possessing neuroprotective action” (Reference: “Phytochemicals as Regulators of Genes Involved in Synucleinopathies”. Biomolecules. 2021; 11(5):624. doi: 10.3390/biom11050624. PMID: 33922207; PMCID: PMC8145209.).

Methods

Line 155 “Cells were grown at 37°C in an atmosphere of 5% CO2/95% air in a cell culture flask.”

The authors should indicate the volume of media and flasks used.  

Lines 198-199: “mRNA from SH-SY5Y differentiated cells were isolated using TRIZOL according to the supplier’s recommendations.” The authors should indicate the supplier.

Discussion

Lines 381-383:”Oxidative stress is a commonality in the pathophysiology of neurodegenerative disorders, such as Alzheimer’s disease, Parkinson’s disease, Huntington’s disease, amyotrophic lateral sclerosis and multiple sclerosis [43] anxiety and depression”

The authors should give the references on the role of oxidative stress in anxiety and depression.

Conclusion and the whole manuscript

“the role of human metabolites from saffron extract”. It is unclear what the authors mean by “human metabolites from saffron extract”

The whole manuscript

The authors use several time the term ”human metabolites from saffron extract” (for example, line 468). The sense of this term is unclear.

Author Response

Reviewer 2

Review of a manuscript “Circulating human serum metabolites derived from the intake of a saffron extract (Safr’insideTM) protect neurons from oxidative stress: consideration for depressive disorders” by Fabien Wauquier and coauthors submitted top “Nutrients” MDPI.

A growing number of mood disorders, including major depressive disorders are a large group of diseases often leading to disability. A significant role in the development of these disorders is played by damage to neurons caused by oxidative stress.  Increased level of oxidative stress is often associated with neurobehavioral alterations, however, the exact mechanism linking them is not completely understood.

The authors developed a robust ex vivo clinical protocol based on human serum enrichment aiming to reveal the connection between activity of human neurons, oxidative stress in depression and the effect of saffron extract on these processes. This is an important area of biomedical research, and the results presented in the manuscript will be interesting for the readership of “Nutrients”.

The following corrections should be made.

Thank you for your compliments. We surely appreciate your positive statements on the robustness and the interest of our clinical ex vivo approach.

Abstract

Line 42-43: “Altogether, these data provide new clinical insights into the mechanisms underlying the beneficial impact of saffron on neuronal viability and activity”. The data presented in the manuscript provide more biochemical, than clinical insights into the mechanisms underlying the beneficial impact of saffron”

We agree and changed the sentence to “Altogether, these data provide new biochemical insights into the mechanisms underlying, in humans, the beneficial impact of saffron on neuronal viability and activity in a context of oxidative stress related to depression.”

Introduction

Lines 83-84: “…and have been reported to promote health-related benefits on emotional well-being [13-16, 21]” . After this sentence the authors should add the following sentence and citation: ”Saffron regulates apoptosis, modulates inflammation and affects mitochondrial dysfunction by upregulating genes, for example, Cyr61, Gpx8, Ndufs4, and Nos1ap possessing neuroprotective action” (Reference: “Phytochemicals as Regulators of Genes Involved in Synucleinopathies”. Biomolecules. 2021; 11(5):624. doi: 10.3390/biom11050624. PMID: 33922207; PMCID: PMC8145209.).

Thanks for this proposition. We added the reference and to avoid overlapping with existing text we adapted the following version: “At a cellular level, saffron was proven, both to regulate apoptosis, modulate inflammatory pathways and affect mitochondrial dysfunction by upregulating gene expression including Cyr61, Gpx8, Ndufs4, and Nos1ap that possess neuroprotective action [22].”

Methods

Line 155 “Cells were grown at 37°C in an atmosphere of 5% CO2/95% air in a cell culture flask.” The authors should indicate the volume of media and flasks used.  

To address your demand we added the information and changed the sentence to: “Cells were grown at 37 °C in an atmosphere of 5% CO2/95% air either in 96 or 24-wells plates with 100µl or 500µl of culture medium respectively.”

Lines 198-199: “mRNA from SH-SY5Y differentiated cells were isolated using TRIZOL according to the supplier’s recommendations.” The authors should indicate the supplier.

We added the information and changed the sentence to : “mRNA from SH-SY5Y differentiated cells were isolated using TRIzol™ Reagent (Ambion – Life Technologies)  according to the supplier’s recommendations.”

Discussion

Lines 381-383:”Oxidative stress is a commonality in the pathophysiology of neurodegenerative disorders, such as Alzheimer’s disease, Parkinson’s disease, Huntington’s disease, amyotrophic lateral sclerosis and multiple sclerosis [43] anxiety and depression”

The authors should give the references on the role of oxidative stress in anxiety and depression.

We gathered and included the references on the role of oxidative stress in anxiety and depression in the text. Line 391: “Oxidative stress is a commonality in the pathophysiology of neurodegenerative disorders, such as Alzheimer’s disease, Parkinson’s disease, Huntington’s disease, amyotrophic lateral sclerosis and multiple sclerosis [44] anxiety and depression [6-9, 45, 46].”

Conclusion and the whole manuscript

“the role of human metabolites from saffron extract”. It is unclear what the authors mean by “human metabolites from saffron extract”

We agree that it may be confusing. In fact, “human metabolites from saffron extract” refers to metabolites found in blood stream following saffron intake in humans. To improve flow and clarity we changed the sentence in the conclusion to : “…we give clues on the role of circulating metabolites produced following saffron extract intake in humans and provide…”

The whole manuscript

The authors use several time the term ”human metabolites from saffron extract” (for example, line 468). The sense of this term is unclear.

According to the aforementioned response we also changed the following sentences:

Line 98: “In this study we combined human metabolism and cell cultures to try and understand whether and how circulating metabolites resulting from saffron extract consumption in humans may influence the activity of human neurons in the context of oxidative stress commonly found in depression.”

Line 332: “Thus, we checked dopamine release in our human neuron cultures. Interestingly, the presence of circulating metabolites from SaE significantly and potently enhanced dopa-mine release (Figure 4), regardless of the presence or absence of oxidative stress.”

Line 348: “Using an original ex vivo clinical approach, we demonstrated that circulating metabolites produced following SaE intake in humans protect human neurons from oxidative stress-induced neurotoxicity by preserving cell viability and BNDF production, while blunting ROS production.”

Line 410: “More importantly, this alteration of the neurons’ activity is prevented in the presence of circulating metabolites of saffron.”

Line 433: “In this study we report that human circulating metabolites resulting from full spectrum saffron extract intake enhanced basal serotonin levels…”

Line 440: “In agreement with these data, we consistently found that circulating metabolites from saffron were able to stimulate both serotonin and dopamine production in human neurons in both presence or even absence of oxidative stress.”

Round 2

Reviewer 1 Report

I would like to thank very much Authors for appropiate correction of the Manuscript.